

# Influence of the sudden stratosphere warming on quasi-2 day waves
Sheng-Yang Gu[1,2,3*], Han-Li Liu[4], Xiankang Dou[1,3], Tao Li[1,3]
[1]CAS Key Laboratory of Geospace Environment, Department of Geophysics and
Planetary Science, University of Science and Technology of China, Hefei, Anhui,
China
[2]Key Laboratory of Earth and Planetary Physics, Institute of Geology and Geophysics,
Chinese Academy of Sciences
[3]Mengcheng National Geophysical Observatory, School of Earth and Space Sciences,
University of Science and Technology of China, Hefei, Anhui, China
[4]High Altitude Observatory, National Center for Atmospheric Research, Boulder,
Colorado, USA
*Corresponding author: S.-Y. Gu, CAS Key Laboratory of Geospace Environment,
School of Earth and Space Science, University of Science and Technology of China,
96 Jin-zhai Rd, Hefei, Anhui 230026, China. (gsy@ustc.edu.cn).



**Abstract:**
The influence of the sudden stratosphere warming (SSW) on quasi-2 day wave
(QTDW) with westward zonal wavenumber 3 (W3) is investigated using the
Thermosphere-Ionosphere-Mesosphere-Electrodynamics General Circulation Model
(TIME-GCM). The summer easterly jet below 90 km is strengthened during an SSW,
which results in a larger refractive index and thus more favorable condition for the
propagation of W3. In the winter hemisphere, the Eliassen Palm (EP) flux diagnostics
indicate that the strong instabilities at middle and high latitudes in the mesopause
region are important for the amplification of W3, which are weakened during SSW
periods due to the deceleration or even reversal of the winter westerly winds.
Nonlinear interactions between the W3 and the wavenumber 1 stationary planetary
wave produce QTDW with westward zonal wavenumber 2 (W2). The meridional
wind perturbations of the W2 peak in the equatorial region, while the zonal wind and
temperature components maximize at middle latitudes. The EP flux diagnostics
indicate that the W2 is capable of propagating upward in both winter and summer
hemispheres, whereas the propagation of W3 is mostly confined to the summer
hemisphere. This characteristic is likely due to the fact that the phase speed of W2 is
larger, and therefore its waveguide has a broader latitudinal extension. The larger
phase speed also makes W2 less vulnerable to dissipation and critical layer filtering
by the background wind when propagating upward.



## 1. Introduction

The westward quasi-2 day wave (QTDW) is a predominant phenomenon in the
mesosphere and lower thermosphere (MLT) region in the summer hemisphere with
zonal wavenumbers 2, 3, and 4. The QTDW was observed by the neutral temperature
measurements from Upper Atmosphere Research Satellite (UARS) [*Wu et al.*, 1996],
Aura satellite [*Tunbridge et al.*, 2011] and Thermosphere Ionosphere and Mesosphere
Electric Dynamics (TIMED) satellite [*Gu et al.*, 2013a], and the neutral wind
measurements from UARS High Resolution Doppler Imager (HRDI) [*Wu et al.*, 1993],
TIMED TIDI [*Gu et al.*, 2013a], and medium frequency radar [*Gu et al.*, 2013b]. In
addition, numerical simulations, including one-dimensional model [*Plumb*, 1983],
two-dimensinoal model [*Rojas and Norton*, 2007], three dimensional TIME-GCM
[*Yue et al.*, 2012] and the Navy Operational Global Atmospheric Prediction System
Advanced Level Physics, High Altitude (NOGAPS-ALPHA) forecast-assimilation
system [*McCormack*, 2009], have also been utilized to study the QTDW. Using
neutral temperature and horizontal wind observations from the TIMED satellite, *Gu et*
*al.* [2013a] showed that the QTDW with westward zonal wavenumber 3 (W3) is
amplified during January/February in the southern hemisphere, and that the QTDW
with westward zonal wavenumber 4 (W4) reaches a maximum amplitude during
July/August in the northern hemisphere. The amplitude of the W3 is nearly twice as
strong as the W4. It is proposed that the W3 is the Rossby-gravity mode (3, 0) [*Salby*,
1981], which can be modulated by the mean flow instabilities [*Plumb*, 1983;
*Limpasuvan et al.*, 2000; *Salby and Callaghan*, 2001; *Yue et al.*, 2012]. The W4 is



first reported by *Rodgers and Prata* [1981] in the radiance data from the Nimbus 6
satellite, which was also confirmed by *Plumb* [1983] with a one-dimensional model
under summer easterly conditions. Usually, the W4 is believed to be an unstable mode
induced by the summer easterly instabilities [*Plumb*, 1983; *Burks and Leovy*, 1986].
Compared with W3 and W4, there are much less reports on the QTDW with westward
zonal wavenumber 2 (W2).
*Tunbridge et al.* [2011] studied the zonal wavenumbers of the summer time
QTDW with satellite temperature observations from 2004 to 2009. They found that
the W2 is amplified mainly during January in the southern hemisphere with a
maximum amplitude at middle latitudes, which always coincides with the temporal
variations of the W3. The horizontal wind observations from the HRDI instrument
onboard the UARS satellite showed that the meridional wind perturbations of the W2
maximize in the equatorial region at the mesopause [*Riggin et al.*, 2004]. This W2
was suggested to be excited in-situ at high altitude, which has little direct connection
with the 2-day activities at lower altitudes. Anomalous 2-day wave activities with
zonal wavenumber 2 were also observed in the Aura/MLS temperature and
line-of-sight wind [*Limpasuvan and Wu*; 2009], which was suggested to be an
unstable mode induced by the strong summer easterly jet during January 2006. *Rojas*
*and Norton* [2007] found a wavenumber 2 westward propagating wave mode with a
period of 49 h in a linear two-dimensional model under boreal summer easterly
condition, which maximized at middle and high latitudes in the summer hemisphere
for both temperature and neutral wind components. The zonal wind and meridional





wind perturbations also exhibited a smaller peak at low latitudes in the winter
hemisphere and at the equator, respectively.

It is known that nonlinear interactions between planetary scale waves can

contribute to atmospheric variability. For example, TIMED/SABER temperature
observations during January 2005 showed that the nonlinear interactions between the
W3 and the migrating diurnal tide could produce an eastward QTDW with zonal
wavenumber 2 [*Palo et al.*, 2007]. The nonlinear interactions between the
quasi-stationary planetary waves (QSPW) and the migrating tides lead to changes in
tides, which then transmit the QSPW signals into the ionosphere at low and middle
latitudes through the E region wind dynamo [*Liu et al.*, 2010; *Liu and Richmond*,
2013]. Nevertheless, the nonlinear interactions between QTDW and other planetary
waves have not been reported.

Rapid growth of QSPWs and their forcing are believed to be the main drivers of

the sudden stratosphere warming (SSW) at high latitudes in the winter hemisphere
[*Matsuno*, 1971], which causes inter-hemispheric connections at different altitudes
[e.g. *Karlsson et al.*, 2007, 2009; *Tan et al.*, 2012]. The wave-mean flow interactions
could decelerate or even reverse the eastward winter stratosphere jet, which, in return,
prevents the further growth of the QSPW. The SSW in the northern hemisphere occurs
usually in January/February, accompanied with a strong zonal wavenumber 1 or 2
QSPW at high latitudes [*Pancheva et al.*, 2008; *Harada et al.*, 2009; *Manney et al.*,
2009; *Funke et al.*, 2010]. There have been recent studies suggesting possible
connection between QTDW and SSW [*McCormick et al.*, 2009; *Chandran et al.*,



2013]. However, it is not clear if this is because both QTDW and SSW tend to occur
in mid to late January, or if the flow condition around SSW is more favorable for
QTDW propagation and/or amplification. In this paper, we investigate the influence
of SSW on QTDW using the National Center for Atmosphere Research (NCAR)
TIME-GCM. The numerial experiments are described in section 2. Section 3 are the
analysis results from the model simulations. Section 4 discusses the contributions of
QTDW to the summer mesospheric polar warming. Our conclusions are presented in
section 5.
**2.  Datasets and analysis**
**2.1  TIMED satellite observations**

The Thermosphere Ionosphere and Mesosphere Electric Dynamics (TIMED)

satellite was launched at the end of 2001, which focuses on the dynamics study of
the mesosphere and lower thermosphere. The TIMED Doppler Imager (TIDI)
instrument on board the TIMED satellite has been providing global horizontal wind
observations since late January 2002. The NCAR-processed version 0307A of P9 line
TIDI wind datasets are utilized here to investigate the inter-annual variations of the
QTDWs during austral summer periods. The vertical resolution of the TIDI winds
between 85 and 105 km is ~2 km, with the highest precision at ~95 km [*Killeen et al.*,
2006]. The version 0307A TIDI horizontal winds have been used in the study of
mesospheric tidal variations and QTDWs [*Wu et al.*, 2008; *Gu et al.*, 2013]. A
two-dimensional least square fitting method, which was provided by *Gu et al.* [2013a;
2015], is also adopted to extract the QTDW signals in this study.



## 2.2 TIME-GCM simulations

The NCAR TIME-GCM simulates the global atmosphere from the upper stratosphere to the thermosphere, and the ionospheric electrodynamics [*Roble and Ridley*, 1994; *Roble*, 2000; *Richmond et al.*, 1992], which is self consistent. The input solar EUV and UV spectral fluxes are parameterized by the solar flux index at 10.7 cm wavelength (F10.7), and it is set to 150 sfu (solar flux unit) in our model simulations. The auroral electron precipitation is parameterized by hemispheric power [*Roble and Ridley*, 1987] and the ionospheric convection is driven by the magnetosphere-ionosphere current system [Heelis *et al.*, 1982]. The hemispheric power is set to 16 and the cross-cap potential is set to 60 in our simulations. The gravity wave forcing is parameterized based on linear saturation theory [*Lindzen*, 1981]. Climatologic migrating tides from the Global Scale Wave Model (GSWM) are specified at the lower boundary. The model is capable of simulating the upward propagation of planetary waves by superimposing periodical geopotential height perturbations at the lower boundary (~30 km). We use the regular horizontal resolution of 5°×5° longitude and latitude grids in the current study. There are 49 pressure levels from 10 hPa (~30 km) to the upper boundary of $3.5 \times 10^{-10}$ hPa (~550 km) with a vertical resolution of one-half scale height. The tides are generally weak compared with climatology in this single version of TIME-GCM. But this does not alter our conclusion with regard to 2-day waves.

To simulate the QTDW, geopotential height perturbations of 1000 m with wavenumber 3 were forced at the TIME-GCM lower boundary. The Gaussian-shaped



149 geopotential height perturbations for W3 peaked at 30°N, extending from 10°S to

150 70°N. To simulate the SSW, geopotential height perturbations of 1000 and 2800 m for

151 a stationary planetary wave with zonal wavenumber 1 (SPW1) were specified at the

152 lower boundary for weak and strong warming, respectively. The Gaussian-shaped

153 geopotential height perturbations for SPW1 peaked at 60°N, extending from 35°N to

154 85°N. In fact, the European Centre for Medium-Range Weather Forecasts (ECMWF)

155 dataset during 2011/2012 austral summer period shows that both the geopotential

156 perturbations of the W3 and SPW1 maximize in the northern (winter) hemisphere at

157 the model lower boundary (not shown). The model was run under perpetual

158 conditions for 40 days with the calendar date set to January 20. Both the W3 and

159 SPW1 gained maximum amplitudes on day 10 with a Gaussian-shaped increase from

160 day 1 to 10. The forcing of W3 was reduced following the same Gaussian function

161 from days 25 to 40. The forcing of SPW1 was sustained from days 10 to 40. The

162 parameters for the control run (base case) and four different experimental runs (case 1,

163 2, 3, and 4) are summarized in Table 1. No W3 or SPW1 forcing was specified at the

164 TIME-GCM lower boundary in the base case, which ran for 15 days to equilibrate and

165 was utilized as initial conditions for the other experimental cases. Case 1 was a

166 standard run for W3 and only geopotential height perturbations of W3 were forced.

167 Case 2 and case 3 were designed to study the amplification of W3 under weak and

168 strong SSW conditions, respectively. The same W3 forcing was added in cases 2 and

169 3, whereas the SPW1 forcing was stronger in case 3 than that in case 2. Case 4 was a

170 standard run for SSW in which only the forcing of SPW1 was included.




## 3. Observational results

Figure 1 shows the ECMWF zonal mean temperature at 80N and 10 hPa from
December to February during 2003-2012. The strongest SSW occurred in January
2009, followed by the second strongest SSW in January 2006. Besides, the SSWs in
2012, 2004 and 2010 were also very strong. Figure 2 shows the temporal variations of
the wave number 3 QTDW in January and February during 2003-2012. The
amplitudes were averaged between 90 and 100 km. The W3 peaked regularly in late
January and early February every year but with strong inter-annual variabilities. For
example, the W3 reached minima in January of 2008 and 2009. It is also clear that the
W3 was strong during the strong SSW years of 2004, 2006 and 2012. Nevertheless,
the W3 was extremely weak during the strongest SSW year of 2009. Figure 3 shows
the averaged amplitudes of the wave number 2 QTDW between 90 and 100 km during
2003-2012, which also maximized in January and February. The W2 was the strongest
during the strong SSW year of 2006, followed by the W2 event in 2012. We can see
that the QTDWs could be very strong during some SSW years, but not during all the
SSW years. Our question is whether the SSW and QTDW (both W2 and W3) impact
each other, and this will be numerically studied in the following section.

## 4.    Simulation results and Discussion

### 4.1 Zonal mean background condition

Since the model time was set perpetually on January 20, the background
temperature and zonal wind in our simulations should show typical northern



winter/southern summer conditions. Figures 4a and 4b show the zonal mean
temperature and zonal mean zonal wind on model day 28 (when W3 peaks) in case 1,
which only has W3 forcing. The zonal mean temperature in TIME-GCM shows a cold
summer mesopause and a warm winter mesopause. The zonal mean zonal wind is
westward in the summer mesosphere and eastward in the winter mesosphere. It is
clear that the global structures of the zonal mean temperature and zonal wind
generally agree with climatology from for example previous TIMED/SABER
temperature [*Mertens et al.*, 2009] and UARS/HRDI wind [*Swinbank and Ortland*,
2003] observations, as well as the NOGAPS-ALPHA forecast assimilations
[*McCormack*, 2009].
We then investigate the atmospheric responses to the weak and strong SSW event
in cases 2 and 3, respectively. Figures 4c and 4e show the temperature differences on
model day 28 between case 2 and case 1, and between case 3 and case 1, respectively.
In cases 1, 2 and 3, the same W3 forcing is specified at the lower boundary, whereas
SPW1 is only specified in cases 2 and 3. The SPW1 forcing in case 2 is weaker than
that in case 3. Compared with case 1, which does not have a stationary planetary wave
specified at the model lower boundary, the temperature of case 2 is warmer by 15-20
K below 60 km and is colder by 20-25 K between 60 and 110 km at high latitudes in
the winter hemisphere. Both the cooling and warming in case 3 are stronger than in
case 2 due to the stronger SPW1 in case 3. The warming and cooling in the
stratosphere and mesosphere for the strong SSW are ~40 K and ~60 K, respectively.
In addition, weaker warming is observed between 70 and 100 km in the middle and





low latitude regions and above 80 km at high latitudes in the summer hemisphere. The
corresponding zonal mean zonal wind differences are shown in Figure 4d and 4f. The
zonal mean zonal wind decreases by ~30 m/s and ~70 m/s in the winter stratosphere
and lower mesosphere in the weak (case 2) and strong (case 3) SSW events,
respectively. It increases by ~30 m/s and ~50 m/s in the mesopause region in the weak
and strong SSW events, respectively. Generally, the SSW features in our simulations
(e.g. the increasing temperature and decreasing westerly in the winter stratosphere
high latitude region) agree with previous reports [*Funke et al.*, 2010; *Yamashita et al.*,
2010; *Tan et al.*, 2012].
**4.2 The influences on W3**
Figure 5a shows the wavenumber-period spectrum of the meridional wind during
days 25-30 of case 1. The meridional wind at ~82 km and 7.5°S is utilized in the
analysis. The westward wavenumber 3 QTDW dominates the whole spectrum, with
negligible signatures at other wavenumbers and periods. The spectra of zonal wind
and temperature show similar W3 signatures as the meridional wind (not shown).
Figure 5b shows the latitudinal and vertical structure of the W3 in meridional wind,
which maximizes at low latitudes in the southern hemisphere mesopause region with
an amplitude of ~60 m/s. Shown in Figure 5c is the structure of the W3 in zonal wind,
which peaks at middle and low latitudes in both hemispheres with maximum
amplitude nearly half of the peak meridional wind amplitude. The zonal wind peak of
~30 m/s in the summer (southern) hemisphere is slightly larger than that of ~20 m/s in
the winter hemisphere, most likely due to the additional amplification by the





baroclinic/barotropic instability of the summer easterly. Figure 5d shows the global
structure of the W3 in temperature, which also peaks at middle latitudes. In the
summer hemisphere, the temperature perturbations peak at ~105 km and ~80 km with
amplitudes of ~7 K and ~8 K, respectively. In the winter hemisphere, the peak of the
W3 at ~80 km is much weaker than that between 100 and 110 km. We should note
that the rapid decay of W3 near the model lower boundary (~30 km) is an artifact near
the model lower boundary. In all, the vertical and latitudinal structures of the 2-day
wave in our simulations generally agree with the TIMED/SABER temperature and
TIMED/TIDI observations [*Palo et al.*, 2007; *Gu et al.*, 2013].

Figure 6 shows the temporal variations of the W3 in meridional wind at ~82 km

for case 1, case 2 and case 3. Note that the same perturbations for W3 were forced at
the lower model boundary for all the three experimental runs. The W3 forcing was
gradually increased from day 1 to 10, and was reduced after day 25 with constant
amplitude between day 10 and 25. The perturbations of SPW1 in case 2 were nearly
three times larger than case 3, both of which were sustained after day 10 with a
Gaussian-shaped increase from day 1 to 10. The W3 in case 1 is the strongest with an
amplitude of ~60 m/s (Figure 6a). The maximum amplitudes of the W3 in case 2 and
case 3 are ~40 m/s and ~35 m/s (Figure 6b and 6c), respectively. It is evident that the
amplitudes of the W3 are weakened during the SSW periods. In the following, we will
examine possible causes of the QTDW decrease during SSW.

The refractive index *m* of a forced planetary wave is [*Andrews et al.*, 1987]:

$$m^2 = \frac{\overline{q}_\varphi}{a(\overline{u}-c)} - \frac{s^2}{(a\cos\varphi)^2} - \frac{f^2}{4N^2H^2} , \qquad (1)$$





where $s$, $c$, $\bar{u}$, $a$, $\varphi$, $f$, $N$, and $H$ are the zonal wavenumber, phase speed, zonal mean
zonal wind, earth radius, latitude, Coriolis parameter, Brunt-Väisällä frequency, and
scale height, respectively. And $\bar{q}_\varphi$ is the latitudinal gradient of the quasi-geostrophic
potential vorticity:
$$\bar{q}_\varphi = 2\Omega\cos\varphi - \left(\frac{(\bar{u}\cos\varphi)_\varphi}{a\cos\varphi}\right)_\varphi - \frac{a}{\rho}\left(\frac{f^2}{N^2}\rho\bar{u}_z\right)_z,$$
(2)

where $\Omega$ is the angular speed of the earth's rotation, $\rho$ is the background air density,
and z means the vertical gradient. A necessary condition for baroclinic/barotropic
instability is $\bar{q}_\varphi < 0$, and the planetary waves are propagating (evanescent) where $m^2$
is positive (negative). Moreover, the meridional and vertical components (EPY and
EPZ) of the Eliassen-Palm (EP) flux vector (F) for planetary waves can also be
calculated with reconstructed wave perturbations from the TIME-GCM, defined
following *Andrews et al.* [1987] as:
$$F = \begin{bmatrix} \text{EPY} \\ \text{EPZ} \end{bmatrix} = \rho a\cos\varphi \begin{bmatrix} \frac{\bar{u}_z\overline{v'\theta'}}{\bar{\theta}_z} - \overline{v'u'} \\ \left[f - \frac{(\bar{u}\cos\varphi)_\varphi}{a\cos\varphi}\right]\frac{\overline{v'\theta'}}{\bar{\theta}_z} - \overline{w'u'} \end{bmatrix}$$
(3)

Here $u'$, $v'$, $w'$ and $\theta'$ are the QTDW perturbations in zonal wind,
meridional wind, vertical wind and potential temperature, respectively.

First, we examine the baroclinic/barotropic instabilities, waveguide and the EP

flux of the W3 for these cases. The averaged zonal mean zonal wind for case 1, case 2
and case 3 during days 25-30, when the W3 reaches the maximum amplitude, are
depicted by the black contour lines in Figures 7a, 7c and 7e, respectively.
Over-plotted are the negative regions of $\bar{q}_\varphi$ by blue shades, which is a prerequisite for





the occurrence of mean flow instability, and the positive regions of the waveguide for
W3 by orange shades, which show where wave progagation is favorable. Shown in
Figures 7b, 7d and 7f are the EP flux vectors (red arrows) of W3 and their divergences
(light blue shades and dot lines) for case 1, case 2 and case 3, respectively. We will
first compare results of case 1 (Figures 7a and 7b) with case 2 (Figures 7c and 7d). A
region of negative $\bar{q}_\varphi$ is seen in case 1 between 80 and 100 km at middle and high
latitudes in the winter hemisphere, which are insignificant in case 2. This difference
probably results from the different vertical shears in zonal wind between the two
cases. Moreover, the region with negative $\bar{q}_\varphi$ in the summer stratosphere polar region
is also slightly more expansive in case 1. Correspondingly, the positive EP flux
divergence for W3, which is an indication of wave source, is stronger in both the
summer mesosphere polar region and the winter mesopause region for case 1. The
positive EP flux divergence near the polar region of summer mesosphere is suggested
to be evidence of wave amplification from the baroclinic/barotropic unstable region
[*Liu et al.*, 2004]. The additional source for the W3 is evident from the positive EP
flux divergence by the southward edge of the baroclinic/barotropic instability in the
winter mesopause region for case 1 (Figure 7b).

Case 1 (Figures 7a and 7b) and case 3 (Figures 7e and 7f) are now compared.

The stratospheric westerlies in the winter hemisphere polar region reverse to easterlies
in case 3, which creates an area with negative $\bar{q}_\varphi$ in the winter polar mesosphere and
stratopause, compared with case 1 (Figures 7a and 7e).   The additional W3 sources
between 60°N and 90°N below 70 km in case 3 may be related to the nearby





instability (Figures 7b and 7f). It is also seen that the summer easterly winds in case 3
are stronger than in case 2 and case 1, which results in a larger refractive index for the
propagation of W3. The EP flux vectors in all the experimental runs show that the W3
propagates mainly southward from the northern hemisphere wave source region at
lower altitudes, and then propagates upward after reaching the southern hemisphere.
These propagation features agree well with previous model simulations [*Chang et al.*,
2011; *Yue et al.*, 2012].

The meridional and vertical components of the W3 EP flux (EPY and EPZ) are

shown in Figure 8. It is clear that both the EPY and EPZ are the strongest in case 1,
which is probably due to the energy transfer to child waves during the nonlinear
interaction between W3 and SPW1 for cases 2 and 3. In the northern (winter)
hemisphere, the stronger EPY and EPZ in case 1 may also be induced by the
additional northern mesospheric baratropic/baraclinic instabilities (shown in Figure
7a), which is not found in case 2 and case 3. The EPY components for all three cases
indicate southward propagation at lower altitudes from the wave source region in the
winter hemisphere, and then northward propagation in the summer polar mesosphere
near the region of instability. The EPZ mostly propagates upward, and is the strongest
at middle and low latitudes in the summer hemisphere and much weaker in the winter
hemisphere. This is in general agreement with the waveguide shown in Figure 7.
Strong upward EPZ at ~30°N and ~100 km is only observed in case 1, which is
probably related to the instability at middle and high latitudes (Figure 7a). Such
instabilities and wave sources disappear in the SSW runs due to the deceleration or





even reversal of the strong winter westerly winds.
Our simulations show that the instabilities at middle and high latitudes in the
winter hemisphere mesopause region can also provide additional and significant
sources for the amplification of W3 (case 1). Such instabilities and the corresponding
sources for W3 are weakened during SSW periods due to the deceleration or even
reversal of the winter stratospheric westerly winds. Our results also show that the
summer easterlies in the stratosphere and lower mesosphere are strengthened during
SSW periods, which results in larger waveguide and thus more favorable background
condition for the propagation of W3. The fact that W3 becomes weaker in the
presence of more favorable propagation conditions (and with the same wave source)
in the summer hemisphere again suggests a loss of W3 wave energy. In the following
section, we argue that the wave energy is transferred to child waves from nonlinear
interaction of W3 with SPW1, namely the QTDW W2 component.
**4.3 Nonlinear interaction between W3 and SPW1**
Figure 9a shows the wavenumber-period spectrum of the meridional wind during
model days 15-20 in case 3 at 100 km and 2.5°N. A westward wavenumber 2 QTDW
dominates the spectrum, which is different from the wavenumber 3 QTDW signature
shown in Figure 5a. The spectra of other components, e.g., zonal wind and
temperature, also show evident wavenumber 2 QTDW signatures. We should
emphasize that W3 and SPW1 are the only planetary waves specified at the lower
boundary of the TIME-GCM and no W2 signals are detected in the TIME-GCM runs
with only W3 or SPW1 perturbations imposed at the lower boundary (case 1 and case





4). Thus, the W2 in case 2 and case 3 is generated by the nonlinear interaction
between W3 and SPW1. The nonlinear interactions between two planetary waves can
generate two child waves with frequencies and zonal wavenumbers being the sum and
difference of the two parent waves [*Teitelbaum and Vial*, 1991]. For the nonlinear
interactions between W3 and SPW1, the frequencies (f, cycles per day) and zonal
wavenumbers (s) of the parents waves are: (f, s) = (0.5, 3) and (0, 1). Note here
positive (negative) s indicates a westward (eastward) propagating wave. Thus the
child waves are: (f, s) = (0.5, 4) and (0.5, 2). However, the wavenumber 4 QTDW is
not well resolved in our simulation due to its lower phase speed and larger dissipation
rate.

Figure 9b shows the cross section of the W2 in meridional wind for case 3 during

model days 15-20. It maximizes in the equatorial and low latitude regions at ~100 km
with a maximum amplitude of ~50 m/s. Shown in Figure 9c is the structure of the W2
in zonal wind and it peaks at middle latitudes with an amplitude nearly half as strong
as the meridional wind. Figure 9d shows the global structure of the W2 in temperature,
which exhibits similar global distributions as zonal wind. The temperature
perturbations show maximum amplitudes of ~10 K in both hemispheres at ~105 km,
and secondary maxima at ~85km: ~7 K in the southern hemisphere and ~5 K in the
northern hemisphere. Figures 10a and 10b show the temporal variations of the W2 in
meridional wind at 100 km for case 2 and case 3, respectively. The perturbations of
the W2 in case 2 are weaker than in case 3, with maximum meridional wind
amplitudes of ~35 m/s and ~55 m/s, respectively. This increase in the W2 amplitude





in case 3 is consistent with the nonlinear interaction mechanism since one of the
parent waves (SPW1) is stronger in case 3, resulting in a stronger child wave.

The mean flow instabilities, the waveguide and the EP flux of W2 are also

examined to study the wave propagation and amplification. Figures 11a and 11c show
the zonal mean zonal wind during model days 15-20, when the W2 reaches the
strongest amplitude, for case 2 and case 3, respectively. In the northern hemisphere,
the mesospheric winter westerlies in case 3 are reversed in the polar region (Figure
11c), resulting in strong instabilities in this region. Weak instabilities are observed at
high latitudes in the winter mesopause region for case 2. In the southern hemisphere,
the summer easterly jet core at middle latitudes is stronger in case 3, which results in
a larger waveguide and thus more favorable condition for the propagation of W2 [*Liu
et al.*, 2004]. The mean flow instabilities in the summer polar region are similar
between case 2 and case 3.
Figures 11b and 11d show the EP flux of W2 and its divergence for case 2 and case 3,
respectively. The EP flux vectors show that W2 propagates in both summer and winter
hemispheres with comparable strength, which accounts for the nearly symmetric
global distribution of the wave perturbations (Figure 9). The propagation features of
W2 are different from W3 on that the W3 is more favorable to propagate in the
summer hemisphere (Figure 7). This is mainly due to the relatively larger phase speed
of W2, which results in a wider latitudinal distribution of positive waveguide for W2
and makes W2 less vulnerable to dissipation and critical layer filtering when
propagating upward in the winter hemisphere [*Salby and Callaghan*, 2001]. Positive





EP flux divergence is seen between 60 and 80 km at middle and high latitudes of the
summer hemisphere for both case 2 and case 3, which is probably due to the wave
amplification by the nearby region of instability [*Liu et al.*, 2004]. In addition, large
positive EP flux divergence regions are found at middle and high latitudes of the
northern hemisphere between 50-100 km for both case 2 and case 3, which is an
indication of wave source due to the nonlinear interaction between SPW1 and W3. In
addition, the positive EP flux divergence of W3 between 30°N and 60°N below 80 km
(Figure 11d) may be related to the negative $\bar{q}_\varphi$ in the winter polar stratosphere (Figure
11c).Figure 12 shows the meridional and vertical components (EPY and EPZ) of the
EP flux of W2 separately. Both the EPY and EPZ are stronger in case 3 than case 2,
which is again consistent with the nonlinear interaction mechanism. The vertical
component EPZ (Figures 12b and 12d) clearly shows that the W2 propagates upward
nearly symmetrically in both summer and winter hemispheres.

Figures 13a and 13b show the EP fluxes of W3 and SPW1 during model days

15-20 in case 3. Strong upward propagating SPW1 from wave source region is seen at
middle and high latitudes in the winter hemisphere. Meanwhile, the energy of W3
propagates mainly southward from the same wave source region. Thus the nonlinear
coupling between SPW1 and W3 is most likely to occur at lower altitudes in the
winter hemisphere near the wave source region. In addition, weaker W3 energy can
also be identified at higher altitudes and at middle and low latitudes in the winter
hemisphere, which, together with the strong SPW1 energy at the same region, could
also contribute to the source of W2 through nonlinear coupling. These speculations





are further investigated by calculating the nonlinear advective tendency between W3
and SPW1. The nonlinear advective tendency terms in the momentum equations,
which have been utilized by *Chang et al.* [2011] in studying the nonlinear coupling
between QTDW and tides, are of the form:
$$\vec{F}_{advection} = -\vec{V} \cdot \nabla V = -\left\{ \frac{u}{a\cos\varphi}\frac{\partial}{\partial\lambda} + \frac{v}{a}\frac{\partial}{\partial\varphi} + w\frac{\partial}{\partial z} \right\} \begin{bmatrix} u \\ v \end{bmatrix}^{T} \qquad (4)$$
Where $u$, $v$ and $w$ are the zonal, meridional and vertical winds, $a$, $z$, $\varphi$ and $\lambda$ are the
earth radius, altitude, latitude, and longitude. By decomposing wind components,
including zonal, meridional and vertical winds, into the forms of $r \approx \bar{r} + r_1 + r_2$ ($\bar{r}$, $r_1$
and $r_2$ represent the zonal mean wind and the wind perturbations of the two planetary
waves, respectively), the zonal and meridional components of the nonlinear coupling
tendencies for two planetary waves are:
$$\vec{F}_{nonlinear,x} = -\frac{1}{a\cos\varphi}(u_1\frac{\partial u_2}{\partial\lambda} + u_2\frac{\partial u_1}{\partial\lambda}) - \frac{1}{a}(v_1\frac{\partial u_2}{\partial\varphi} + v_2\frac{\partial u_1}{\partial\varphi}) - (w_1\frac{\partial u_2}{\partial z} + w_2\frac{\partial u_1}{\partial z}) \quad (5)$$
$$\vec{F}_{nonlinear,y} = -\frac{1}{a\cos\varphi}(u_1\frac{\partial v_2}{\partial\lambda} + u_2\frac{\partial v_1}{\partial\lambda}) - \frac{1}{a}(v_1\frac{\partial v_2}{\partial\varphi} + v_2\frac{\partial v_1}{\partial\varphi}) - (w_1\frac{\partial v_2}{\partial z} + w_2\frac{\partial v_1}{\partial z}) \quad (6)$$
where $\bar{u}$, $\bar{v}$ and $\bar{w}$ are the zonal mean zonal, meridional and vertical winds, $u_1$
and $u_2$, $v_1$ and $v_2$, $w_1$ and $w_2$ are the zonal, meridional, vertical wind perturbations for
two different planetary waves. By adopting a complex perturbation of the form
$u' = \hat{u}e^{i(\sigma t - s\lambda)}$ (the $\sigma$ and $s$ are the frequency and zonal wavenumber of the planetary
wave, $t$ is the universal time), the complex amplitudes of the nonlinear advective
tendencies can be calculated as:
$$\vec{F}_{nonlinear,x} = \frac{i\hat{u}_1\hat{u}_2}{a\cos\varphi}(s_1 + s_2) - \frac{1}{a}(\hat{v}_1\frac{\partial\hat{u}_2}{\partial\varphi} + \hat{v}_2\frac{\partial\hat{u}_1}{\partial\varphi}) - (\hat{w}_1\frac{\partial\hat{u}_2}{\partial z} + \hat{w}_2\frac{\partial\hat{u}_1}{\partial z}) \qquad (7)$$



$\vec{F}_{nonlinear,y} = \frac{i}{a\cos\varphi}(\hat{u}_1\hat{v}_2 s_2 + \hat{u}_2\hat{v}_1 s_1) - \frac{1}{a}(\hat{v}_1\frac{\partial\hat{v}_2}{\partial\varphi} + \hat{v}_2\frac{\partial\hat{v}_1}{\partial\varphi}) - (\hat{w}_1\frac{\partial\hat{v}_2}{\partial z} + \hat{w}_2\frac{\partial\hat{v}_1}{\partial z})$  (8)
where $s_1$ and $s_2$ are the zonal wavenumbers of different planetary waves, $\hat{u}_1$
and $\hat{u}_2$, $\hat{v}_1$ and $\hat{v}_2$, $\hat{w}_1$ and $\hat{w}_2$ are the zonal, meridional, vertical wind
amplitudes for two different planetary waves.

Figure 13c shows the amplitude of the meridional component of the nonlinear

advective tendency between W3 and SPW1 (equation 8). The nonlinear coupling
between W3 and SPW1 maximizes at lower altitudes in the northern hemisphere,
which is not surprising since both the W3 and SPW1 perturbations are forced at the
lower model boundary in the northern hemisphere. Correspondingly, a strong W2
source is present at lower altitudes in the northern hemisphere, which is also
suggested by the positive EP flux divergence shown in Figure 11d. The large
nonlinear advection value at the lower boundary is due to the large wave sources
forced there to compensate for the unrealistic wave decay usually found near the
model lower boundary. Although the amplitude of the advective tendency at the lower
model boundary may be too large, it is still likely that the nonlinear interaction
between W3 and SPW1 at ~10 hPa in the winter hemisphere is strong, since
climatologically the sources of W3 and SPW1 are found to maximize in the winter
hemisphere at stratospheric heights. There is an additional region extending from 60
km to about 100 km at low to mid latitudes where the advective tendency term
becomes significant (with a peak at ~70km). This is again consistent with the positive
EP flux divergence in Figure 11d, and is likely due to the nonlinear coupling of W3
and SPW1.





**5. Conclusions**

The influence of the SSW on the QTDW was investigated with NCAR

TIME-GCM simulations. The westward wavenumber 3 QTDW was simulated by
specifying geopotential height perturbations of 1000 m at the lower model boundary
(~30 km) for both the standard W3 run and the SSW runs. Wavenumber 1 stationary
planetary waves with geopotential height perturbations of 1000 m and 2800 m were
forced in the northern hemisphere at the lower model boundary to induce minor and
major SSWs, respectively.

We find that the mean flow instabilities at middle and high latitudes in the winter

mesopause region can provide additional and essential sources for the amplification of
W3, whereas such instabilities are weakened during SSW periods due to the
deceleration or even reversal of the winter westerlies. The mean flow instabilities in
the winter stratosphere polar region, induced by the mean wind reversal from westerly
to easterly during SSW periods may also contribute to the amplification of W3. The
waveguide of the W3 is larger during SSW periods, which favors the propagation of
W3. The wave energy of W3 could be transmitted to child waves through the
nonlinear interaction between W3 and stationary planetary waves during the SSW
periods.

The nonlinear interaction between W3 and the SPW1 results in a new kind of

westward QTDW with zonal wavenumber 2. The W2 is generated mainly in the wave
source region, and then propagates into both summer and winter hemispheres. The
meridional wind perturbations of W2 maximize in the equatorial region, whereas the





zonal wind and temperature components peak at middle latitudes. The EP flux
diagnostics show that W2 is capable of propagating in both hemispheres, which
results in much more symmetric global structures than W3 for both wind and
temperature components. This is probably due to the larger phase speed of W2, which
results in larger latitudinal distributions of positive waveguide and makes W2 less
vulnerable to dissipation and critical layer filtering by the background wind when
propagating upward. In the summer hemisphere, the instabilities in the upper
stratosphere and lower mesosphere polar region may contribute to the amplification of
W2 through wave-mean flow interaction. In the winter hemisphere, the nonlinear
coupling between W3 and SPW1 at middle and low latitudes between 50 km and 100
km, and the instabilities induced by the reversal of winter stratosphere westerly during
SSW periods, most probably provide additional sources for W2. The stronger
stationary planetary wave accounts for the stronger W2 perturbations during major
SSW period by transmitting more energy to W2 during the nonlinear interaction
between W3 and SPW1. Moreover, the background mean flow condition is also more
favorable for the propagation of W2 during major SSW period with a larger
waveguide. We should note that the amplitudes of W3 and SPW1 specified at the
lower boundary were both set to constant values in our simulation, while the wave
sources would vary with time in real atmosphere. In the future, we plan to use more
realistic assimilation datasets (e.g., ECMWF) as the lower model boundary to further
study the influence of SSW on QTDWs, to understand the variability of the wave
sources, and their possible relation with SSW.




## Acknowledgements

This work is funded by the Project Supported by the Specialized Research Fund for
State Key Laboratories, the Project Funded by China Postdoctoral Science Foundation,
the National Natural Science Foundation of China (41274150, 41421063), the
Chinese Academy of Sciences Key Research Program (KZZD-EW-01-1), the National
Basic Research Program of China (2012CB825605). The data utilized in this paper is
from TIME-GCM simulations on NCAR Yellowstone computing system
(ark:/85065/d7wd3xhc), sponsored by the National Science Foundation. H.L.
acknowledges support from NSF grant AGS-1138784.



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






| | GP Height of W3 | GP Height of SPW1 |
|---|---|---|
| Base case | × | × |
| Case 1 | 1000 m | × |
| Case 2 | 1000 m | 1000 m |
| Case 3 | 1000 m | 2800 m |
| Case 4 | × | 2800 m |

**Table 1.** The geopotential height perturbations of W3 and SPW1 specified at the
lower model boundary for different model runs.










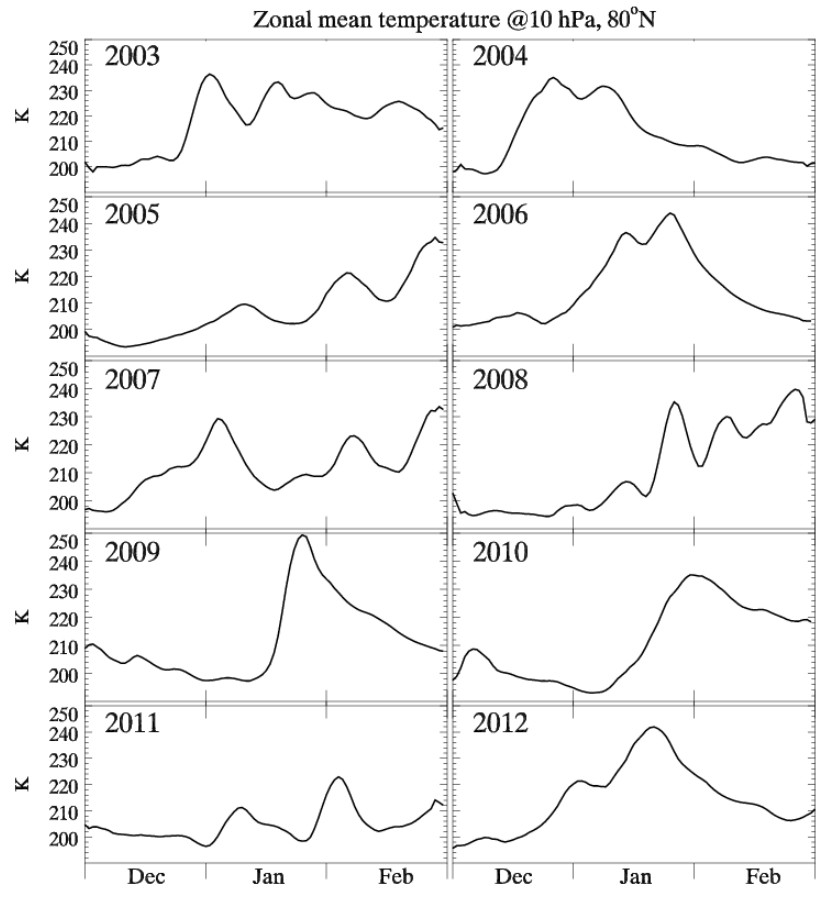


Figure 1. The ECMWF zonal mean temperature at 80°N and 10 hPa from December
to February during 2003-2012.





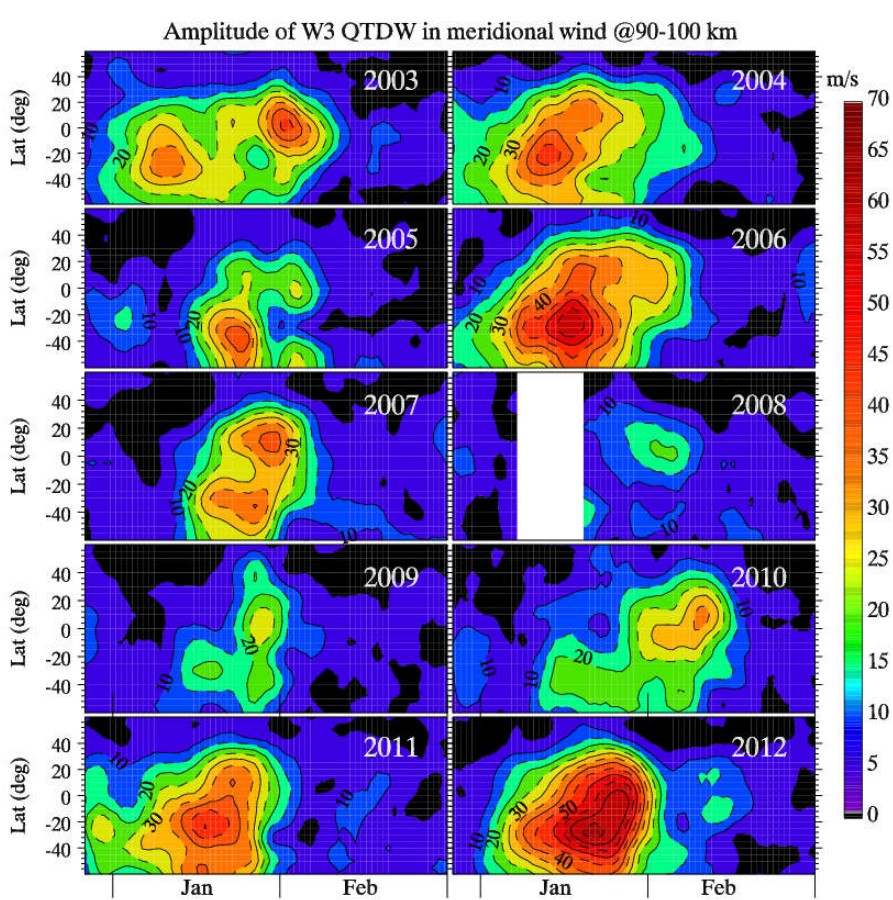


Figure 2. The temporal variations of the wave number 3 QTDW in January and
February during 2003-2012. The amplitudes are averaged between 90 and 100 km.





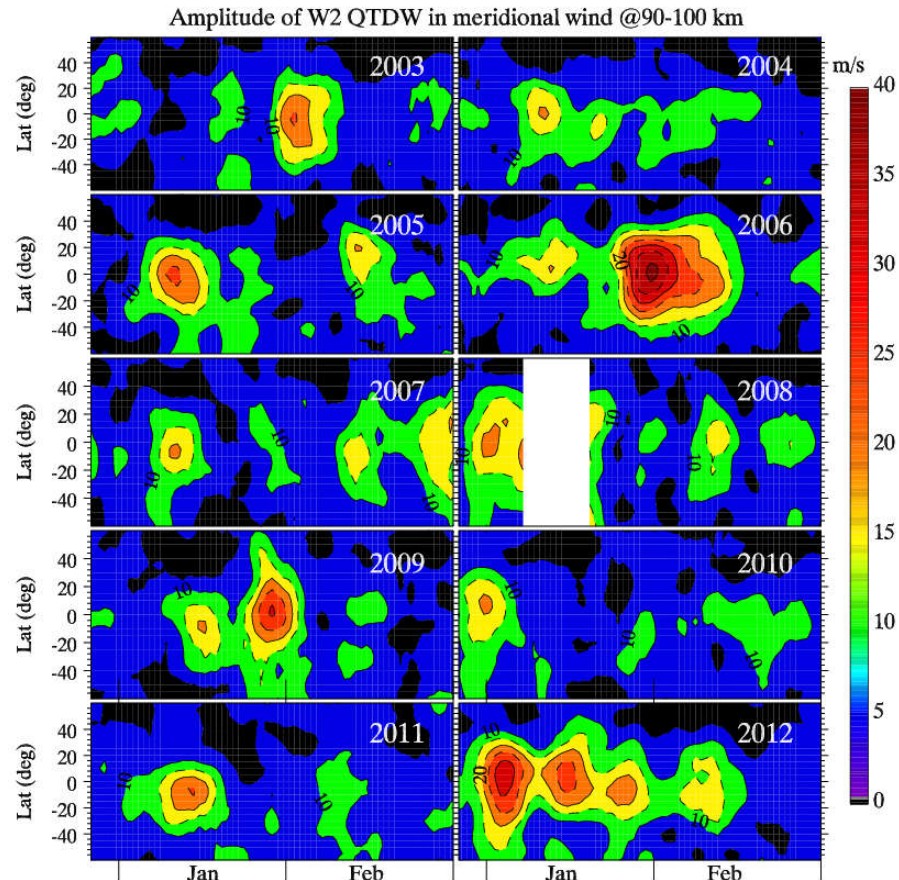


Figure 3. The same as Figure 2 but for the wave number 2 QTDW.




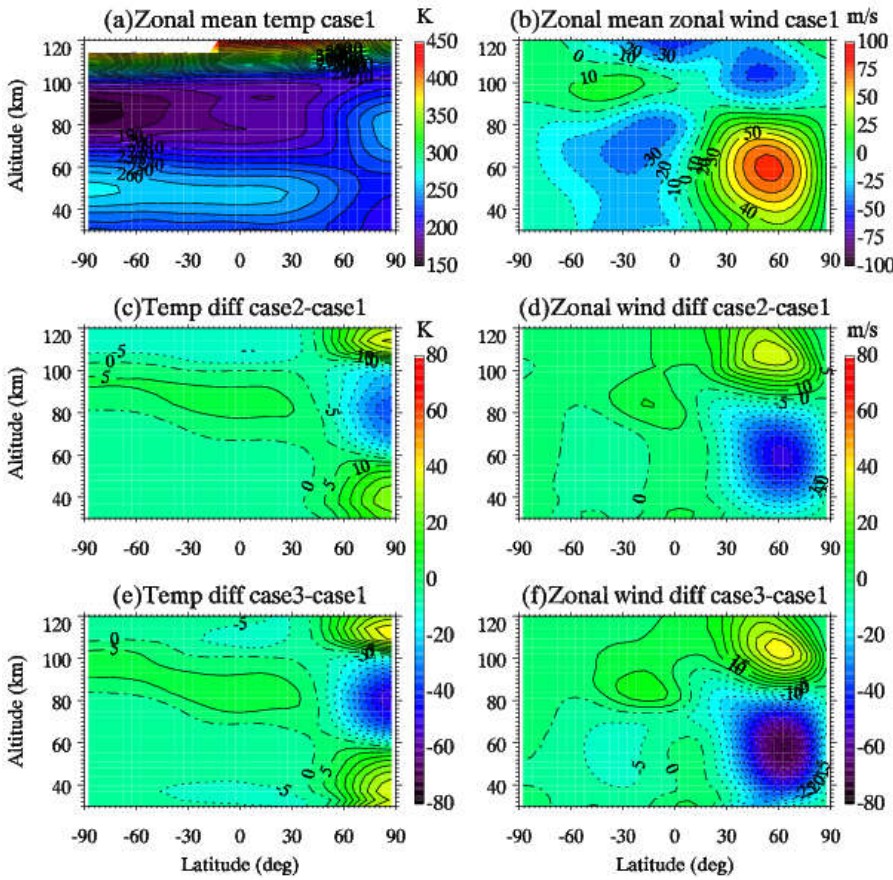


**Figure 4.** The zonal mean (a) temperature and (b) zonal wind in case 1 on model day
28. The temperature and zonal wind differences between (c, d) case 2 and case 1, (e, f)
case 3 and case 1 are also shown. The temperature contour intervals are 10 K in (a)
and 5 K in (c) and (e). The zonal wind contour intervals are 10 m/s in (b) and 5 m/s in
(d) and (f).



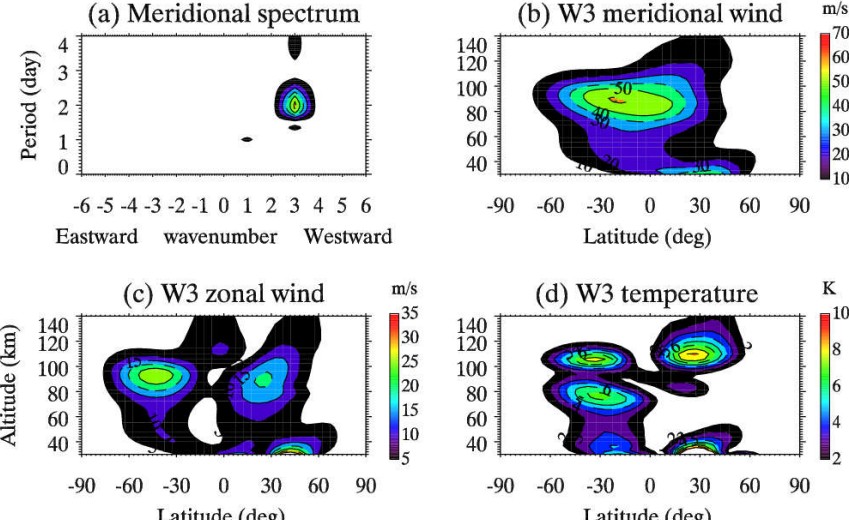

**Figure 5.** (a) The least-square fitting spectrum of the meridional wind at 22.5°S and ~90 km during model day 25-30 of case 1. A westward wave number 3 QTDW dominates the spectrum. The vertical and global structures of the W3 in meridional wind, zonal wind and temperature are shown in (b), (c) and (d), respectively. The contour intervals are 10 m/s, 5 m/s and 1 K for meridional wind, zonal wind and temperature, respectively.





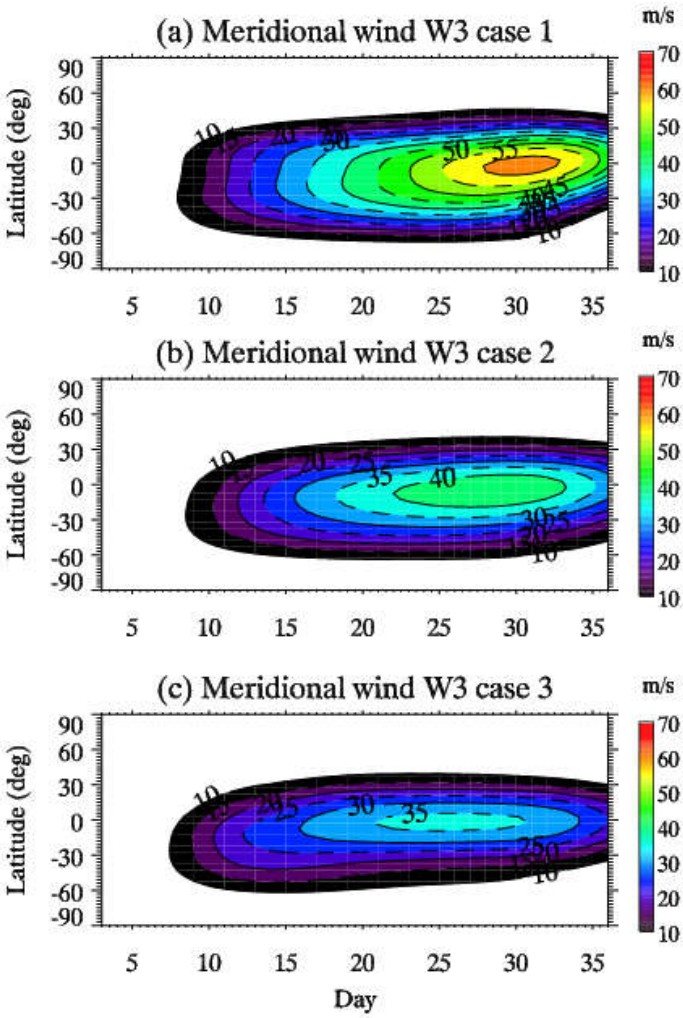

**Figure 6.** The temporal variations of the W3 at 82 km for (a) case 1, (b) case 2 and (c)
case 3. Geopotential height perturbations of 1000 m are forced at the lower boundary
for all the three control runs to simulate the W3. SPW1 geopotential height
perturbations of 1000 m and 2800 m are forced at the lower boundary to induce the
weak and strong SSWs in case 2 and case 3, respectively. No SPW1 perturbations are
forced at the lower boundary of case 1. The contour intervals are 5 m/s.





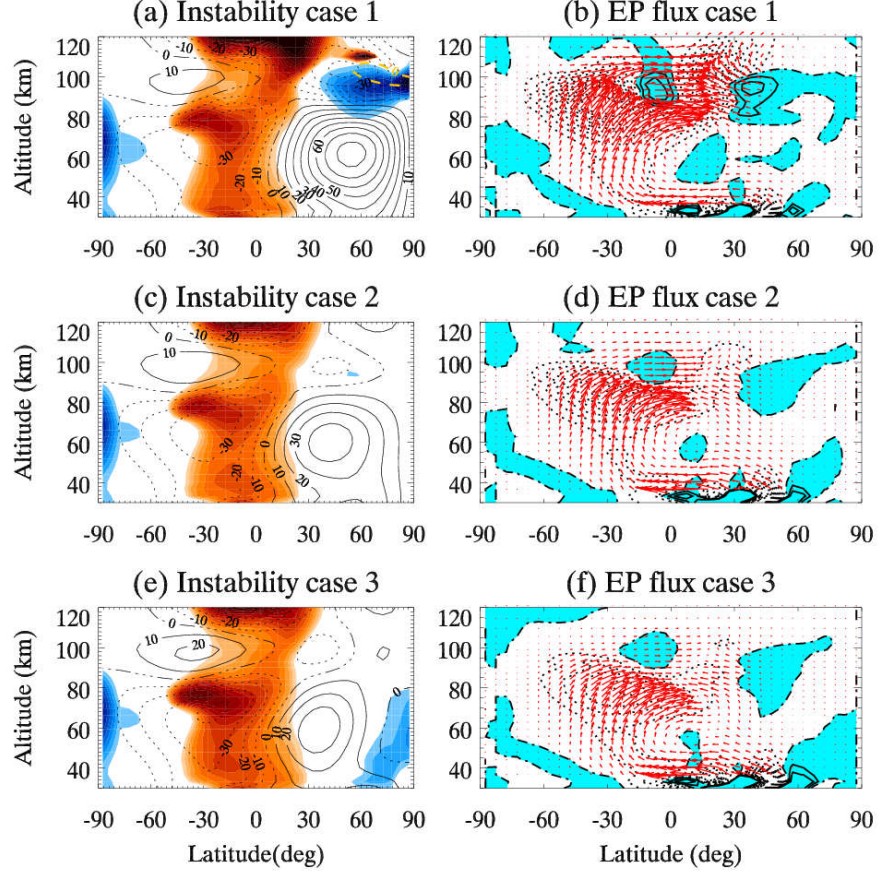

**Figure 7.** The zonal mean zonal wind during model days 25-30 for (a) case 1, (c) case
2 and (e) case 3. The baroclinic/barotropic instabilities are overplotted with blue
shades. The orange shaded region denotes the positive (propagating) waveguide ($m^2$)
for W3. Shown on the right are the EP flux vectors (red arrows) and their divergences
(light blue shade for positive value, dot line for negative value) for (b) case 1, (d) case
2 and (f) case 3. The contour intervals for the EP flux divergence are 2 m/s/day.



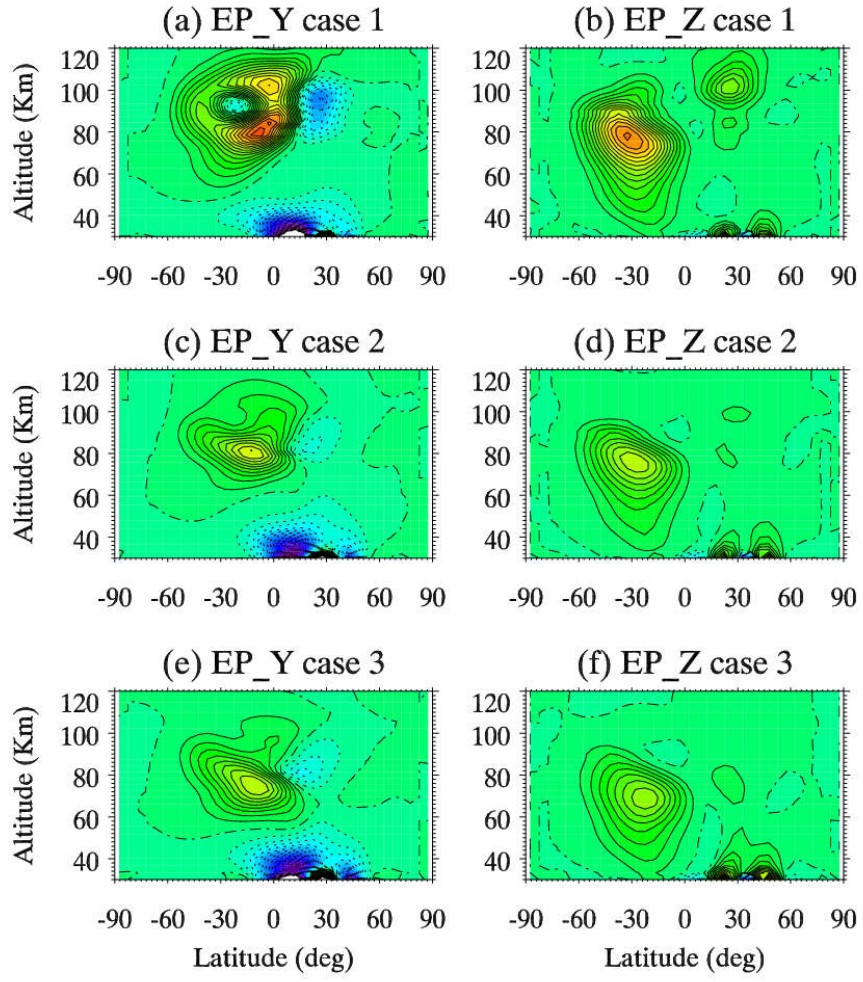

**Figure 8.** (left) Meridional and (right) vertical components of the EP flux of the W3
during model day 25-30 for (a, b) case 1, (c, d) case 2 and (e, f) case 3. The solid
contours are for northward or upward directions. Both components have been
normalized by the air density.





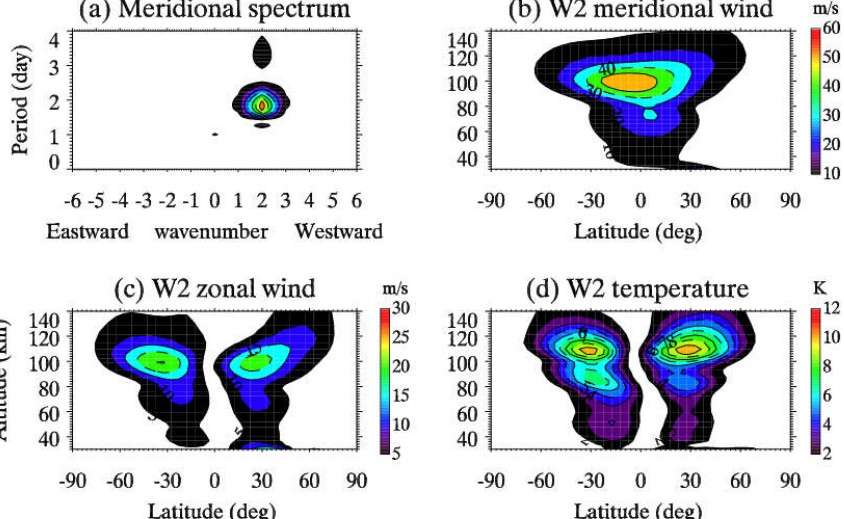


**Figure 9.** Similar to Figure 5 but for case 3 during model days 15-20. Figure 9a
shows the meridional wind spectrum at 100 km and 2.5°N. Figures 9b, 9c and 9d
show the global and vertical structures of W2 for meridional wind, zonal wind and
temperature, respectively.




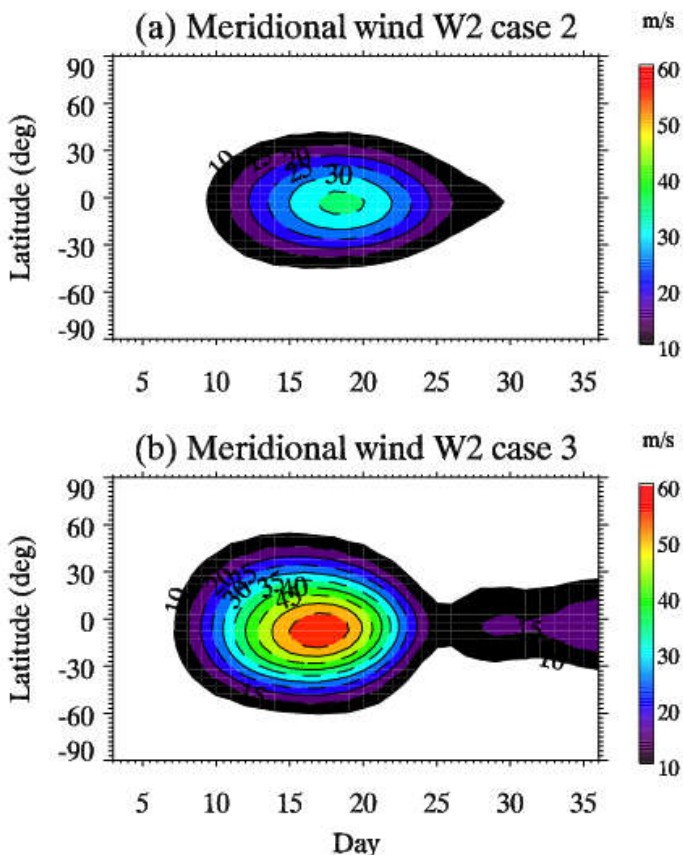

**Figure 10.** The temporal variaitions of the W2 at 100 km for (a) case 2 and (b) case 3.
The contour intervals are 5 m/s.






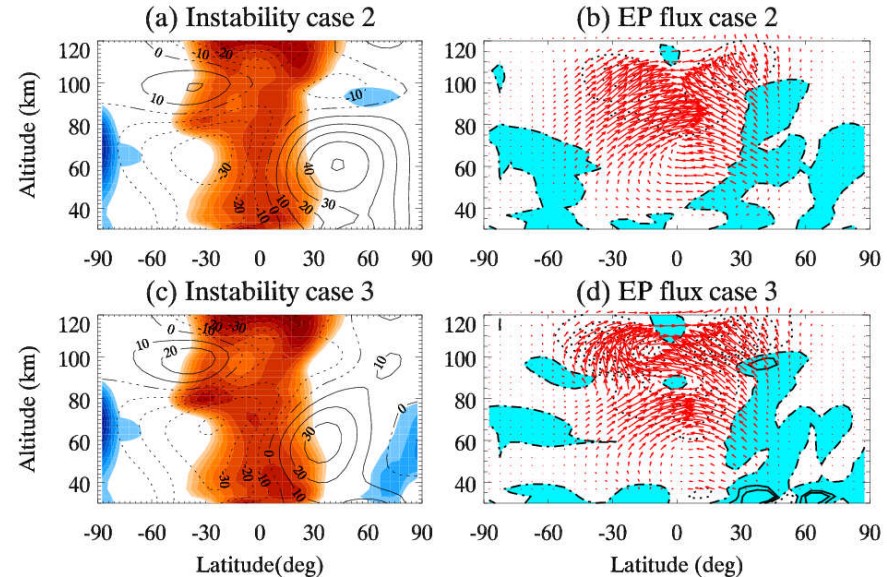

**Figure 11.** The same as Figure 7 but for the W2 during model day 15-20 for (a, b) case 2 and (c, d) case 3.





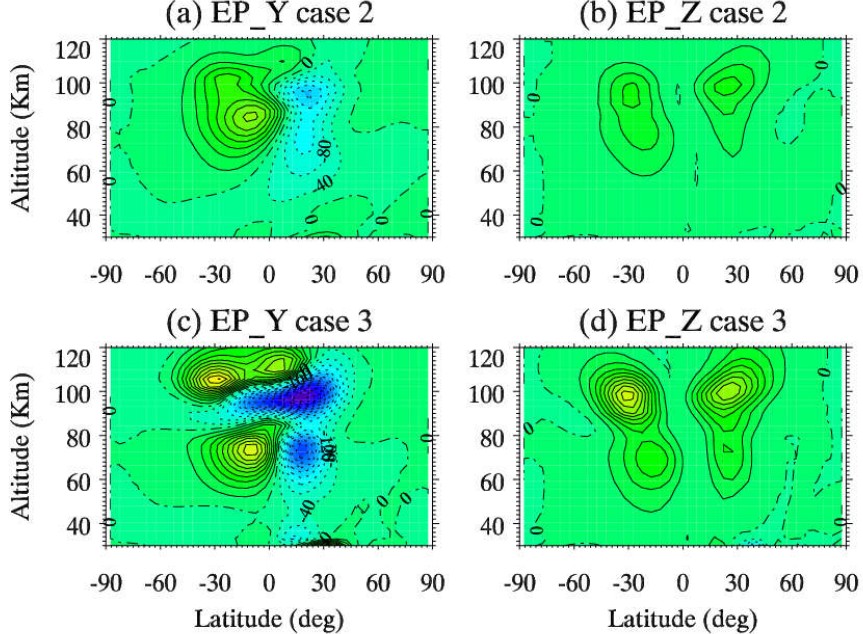

**Figure 12.** The same as Figure 8 but for the W2 during model day 15-20 for (a, b)
case 2 and (c, d) case 3.



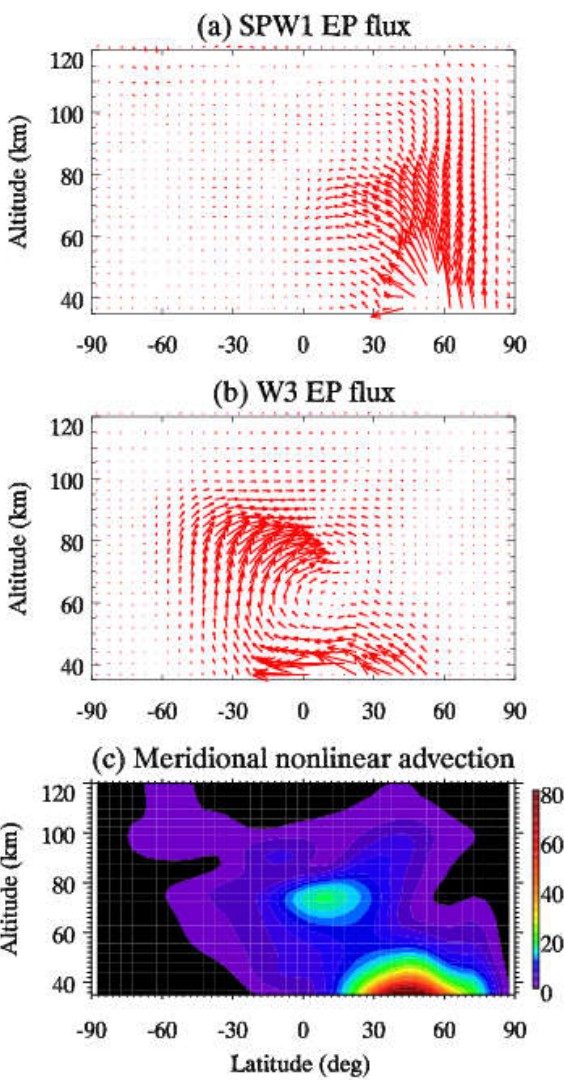


**Figure 13.** The EP flux vectors of (a) the SPW1 and (b) the W3 during model day
15-20 of case 3. (c) The amplitude (m/s$^2$) of the meridional component of the
nonlinear advection tendency between W3 and SPW1.