# Peer review of "Manuscript under review for journal Atmos. Chem. Phys."

_Atmospheric Chemistry and Physics, 2015_

## Referee Comment (RC1) · Anonymous Referee #1 · 4 Mar 2016

GENERAL COMMENTS

The paper is an interesting contribution on the appearance of QTDW. TIME-GCM simulations are used to separate the impact of forcing of planetary waves at the surface. Although the simulated SSW is only minor, the data base is evaluated to show an overall decrease of westward PW3 due to reduced instabilities and an increase of westward PW2 due to nonlinear interactions. I suggest an extended discussion of these results with respect to observations and recommmend: minor revision.

SPECIFIC COMMENTS

1) Mesospheric instabilities: The simulations bring up a SSW and are sufficient to discuss certain pathways leading to QTDWs. However, the warming is minor and possibly not sufficiently strong to explain observations of major warmings. One of the obvious

effects is the missing of instabilities in the stratospheric easterlies during these times. Its potential to generate planetary waves is lined out in Liu et al. (2004), Limpasuvan et al. (2012), Zülicke & Becker (2014) and Sato & Nomoto (2015), for example.

2) Equatorial instabilities: It should at least be mentioned, that the intensifying equatorial stratopause easterlies may also lead to instabilities and subsequent forcing of QTDW (Limpasuvan et al., 2000).

3) Observations: For the conclusion of the paper I would like the authors to add a discussion observations in the context of W2 and W3 relation to SSWs. Beside of the relatively weak SSW modelled here, it could also be that a SPW2 forcing (a split-vortex SSW) may directly lead to stronger QTDW.

TECHNICAL CORRECTIONS

With respect to common use of wordings, I suggest the following: a) use "sudden stratospheric warming" instead of "sudden stratosphere warming" b) use "nonlinear advection" instead of "nonlinear advetive"

In the following, all numbering refers to the discussion paper "acp-2015-982.pdf"

line 49: "TIME-GCM" should be defined.

line 86: "TIMED/SABER" should be defined.

line 96: stratospherIC - see a) above.

line 115: If defined before as sugested, "TIMED" need not be defined here again.

line 197: Here it seems to me that "eastward" and "westward" were confused.

line 261: I suggest to start the sentence not with "And" but "Further, " for example.

line 271: If you want to indicate vectors with an over-arrow as you do later, I suggest to do this here, too. Also, because you later introduce another flux, I would add here the index "EP".

line 288: "expansive" –> "extended"

line 294: "by" –> "at"

line 273: The sentence "In the northern... region" confused me. I see in your plots that the mesospheric winter easterlies (!) reversed, resulting in weak (!) instabilities in this region. Please, reconsider the text.

line 409: You write of "strong" SPW1 energy, while I see in fig. 13a in 0 - 30 °N at 60 - 80 km only moderate SPW1 fluxes in comparison to the stronger fluxes at about 60 °N. Please, clarify.

line 411: "advective" –> "advection", as done in the figure captions - see b) above.

line 415: Add an arrow over the "V" after the "nabla".

line 422: could be deleted because not used.

line 423: Delete arrow because it is a vector component only.

line 426 - 434: In order to save place for additional discussion, I suggest to delete the text "By adopting... waves." This is for my taste only technical information which does not need to be explained.

line 445: You write the amplitude may be "too large" - please, explain why? What did you take for reference?

line 446: Please, add the corresponding kilometers, which is the unit of the vertical axes.

line 450: Although this peak in Fig. 13c is not one-to-one at the same position as the one in Fig. 11d, I follow your argument.

line 466: This is right, and this is what I mean with my specific comment 1). Only the present simulations do not show this instability because the SSW is too weak. Please, mention this because it is important when discussing oservations.

REFERENCES

Becker, E., 2012: Dynamical control of the middle atmosphere. Space Sci. Rev. 168: 283 - 314. doi:10.1007/s11214-011-9841-5.

Limpasuvan, V., C. B. Leovy, Y. J. Orsolini & B. A. Boville, 2000: A numerical simulation of the two-day wave near the stratopause. J. Atmos. Sci. 57: 1702 - 1717.

Limpasuvan, V., J. H. Richter, Y. J. Orsolini, F. Stordal & O.-K. Kvissel, 2012: The roles of planetary and gravity waves during a major stratospheric sudden warming as characterized in WACCM. J. Atmos. Sol.-Terr. Phys. 78-79: 84 - 98. doi:10.1016/j.jastp.2011.03.004.

Sato, K. & M. Nomoto, 2015: Gravity Wave–Induced Anomalous Potential Vorticity Gradient Generating Planetary Waves in the Winter Mesosphere. J. Atmos. Sci. 72, 9: 3609-3624. doi:10.1175/jas-d-15-0046.1.

Zülicke, C. & E. Becker, 2013: The structure of the mesosphere during sudden stratospheric warmings in a global circulation model. J. Geophys. Res. Atmos. 118: 2255 - 2271. doi:10.1002/jgrd.50219.
* * *

---

## Referee Comment (RC2) · Anonymous Referee #2 · 7 Mar 2016

General comments:

The authors use the thermosphere-Ionosphere-Mesosphere-Electrodynamics General Circulation model to model the effect of the sudden stratospheric warming on the quasi-2- day wave (QTDW). They investigate the non-linear interaction of the QTDW with westward zonal wavenumber 3 (W3) and the stationary planetary wave with zonal wavenumber 1 and show that a QTDW with westward zonal wavenumber 2 can be produced.

Specific comments:

I agree with the specific comments of referee nr. 1 in addition I was wondering:

1) Why has the analysis of the W3 wave been performed in the meridional wind at 82

km, 7.5 S and days 25-30

2) Why are days 15-20 chosen for the analysis of the W2 and not the same time period as for the analysis of the W3?

3) Why has the analysis of the W2 wave been performed in the meridional wind at 100 km, 2.5 N and during days 15-20?

4) In the caption of Figure 5 the authors state that the analysis of the W3 has been performed at 22.5S and ~90km. However, in the text describing Figure 5 ( page 11, line 226f), the authors state that the analysis of the W3 wave has been performed at ~82 km and at 7.5S. Which coordinates have been used?

Technical corrections:

1) Page 15, Line 313: Baratropic/baraclinic => barotropic/baroclinic

---

## Author Comment (AC1) · 15 Mar 2016

Interactive comment on "Influence of the suddenstratosphere warming on quasi-2 day waves" byS.-Y.Gu et al. Anonymous Referee #1

GENERAL COMMENTS The paper is an interesting contribution on the appearance of QTDW. TIME-GCM simulations are used to separate the impact of forcing of planetary waves at the surface. Although the simulated SSW is only minor, the data base is evaluated to show an overall decrease of westward PW3 due to reduced instabilities and an increase of westwardPW2 due to nonlinear interactions. I suggest an extended discussion of these results with respect to observations and recommmend: minor revision.

[Figure]

Reply: We thank the reviewer for reading through the manuscript and the following are responses to all the comments/suggestions made by the reviewers, and the tracked version is also attached for further interactive discussions.

SPECIFIC COMMENTS 1) Mesospheric instabilities: The simulations bring up a SSW and are sufficient to discuss certain pathways leading to QTDWs. However, the warming is minor and possibly not sufficiently strong to explain observations of major warmings. One of the obvious effects is the missing of instabilities in the stratospheric easterlies during these times. Its potential to generate planetary waves is lined out in Liu et al. (2004), Limpasuvanet al. (2012), Zülicke& Becker (2014) and Sato &Nomoto (2015), for example.

Reply: In our TIME-GCM simulations, the SSW is generated by forcing stationary planetary waves with zonal wave number (SPW1) at the lower model boundary ($\sim$10 hPa or 30 km). An amplitude of 2.8 km for SPW1 is set in case 3 to generate the strong SSW event in our simulation. Although case 3 may not be a major SSW according to standard definition (and this is a limitation of the model configuration with its lower boundary specified at 10hPa), it does cause strong jet reversal (from westerly to easterly) above $\sim$40km. As shown in Figure 7(e), there is indeed a region with negative potential vorticity gradient corresponding to this reversal. In the case of a major warming, the exact altitude of the instability may be different, but the process should be qualitatively similar to the results presented here. We have revised the paper to reflect this point, and added the references suggested.

2) Equatorial instabilities: It should at least be mentioned, that the intensifying equatorial stratopause easterlies may also lead to instabilities and subsequent forcing of QTDW (Limpasuvan et al., 2000).

Reply: We note that Limpasuvan et al. [2000] found the inertial instability could play a role in amplifying QTDW. Nevertheless, the TIME-GCM experiments performed by Liu et al. [2004] showed that the inertial instability does not seem to greatly enhance the

wave response but only causes additional spatial variability in the equatorial region. This is mentioned in the revision.

3) Observations: For the conclusion of the paper I would like the authors to add a discussion observations in the context of W2 and W3 relation to SSWs. Beside of the relatively weak SSW modelled here, it could also be that a SPW2 forcing (a split-vortex SSW) may directly lead to stronger QTDW.

Reply: Following the reviewer's comment, we added discussions in the conclusion regarding QTDWs from TIMED observation. We agree that the SPW2 forcing may provide additional variabilities for QTDW, but this may not necessarily mean stronger QTDW. Since the growth of QTDW is sensitive to the background wind, and only the modest mean wind is favorable for the propagation and amplification of QTDW. Nevertheless, it will be interesting in future studies to compare the QTDW activities during split-vortex and displacement-vortex years (e.g., 2006 and 2009), which may show new light on the inter-annual variability of QTDW.

TECHNICAL CORRECTIONS With respect to common use of wordings, I suggest the following: a) use "sudden stratospheric warming" instead of "sudden stratosphere warming" b) use "nonlinear advection" instead of "nonlinear advective"

Reply: The words are corrected in the revision.

In the following, all numbering refers to the discussion paper "acp-2015-982.pdf"

line 49: "TIME-GCM" should be defined.

Reply: TIME-GCM is defined in the revision.

line 86: "TIMED/SABER" should be defined.

Reply: It is defined in the revision.

line 96: stratospherIC - see a) above.

Reply: It is changed in the revision.

line 115: If defined before as suggested, "TIMED" need not be defined here again.

Reply: The definition of TIMED is deleted here.

line 197: Here it seems to me that "eastward" and "westward" were confused.

Reply: The summer easterly in propagates from the east to the west, and the winter westerly propagates from the west to the east. Thus, we use westward and eastward instead of easterly and westerly in our manuscript. The sentence is revised as "In the upper stratosphere and mesosphere, the zonal mean zonal wind is easterly in the summer hemisphere and westerly in the winter hemisphere."

line 261: I suggest to start the sentence not with "And" but "Further, " for example.

Reply: "And" is replaced by "Besides" in the revision.

line 271: If you want to indicate vectors with an over-arrow as you do later, I suggest to do this here, too. Also, because you later introduce another flux, I would add here the index "EP".

Reply: Added in the revision.

line 288: "expansive" –> "extended"

Reply: Changed in the revision.

line 294: "by" –> "at"

Reply: Changed in the revision.

line 273: The sentence "In the northern... region" confused me. I see in your plots that the mesospheric winter easterlies (!) reversed, resulting in weak (!) instabilities in this region. Please, reconsider the text.

Reply: We think the reviewer means line 373 here. The instabilities are indeed not

strong. The sentence is revised to be "In the northern hemisphere of case 3, the eastward wind in the upper stratosphere and lower mesosphere reverses in the polar region (Figure 11c), resulting in weak instabilities in this region."

line 409: You write of "strong" SPW1 energy, while I see in fig. 13a in 0 - 30 âŮęN at 60-80 km only moderate SPW1 fluxes in comparison to the stronger fluxes at about 60âŮęN. Please, clarify.

Reply: We mean that SPW1 energy is still much stronger in 0-30âŮęN at 60-80 km than that in the southern (summer) hemisphere. The word "strong" is removed in the revision to avoid confusions.

line 411: "advective" –> "advection", as done in the figure captions - see b) above.

Reply: Changed in the revision.

line 415: Add an arrow over the "V" after the "nabla".

Reply: Added in the revision.

line 422: could be deleted because not used.

Reply: Deleted in the revision.

line 423: Delete arrow because it is a vector component only.

Reply: Deleted in the revision.

line 426 - 434: In order to save place for additional discussion, I suggest to delete the text "By adopting... waves." This is for my taste only technical information which does not need to be explained.

Reply: Deleted in the revision.

line 445: You write the amplitude may be "too large" - please, explain why? What did you take for reference?

[Figure]

Reply: We mean the nonlinear advection at lower boundary is much larger than the peak in the mesosphere. The large nonlinear advection value at the lower boundary is due to the large wave sources forced there to compensate for the unrealistic wave decay usually found near the model lower boundary. This is pointed out in the text. WACCM, which simulates the atmosphere from the ground, will be used in the future to avoid the lower boundary effect.

line 446: Please, add the corresponding kilometers, which is the unit of the vertical axes.

Reply: We changed ∼10 hPa with ∼30-45 km in the revision.

line 450: Although this peak in Fig. 13c is not one-to-one at the same position as the one in Fig. 11d, I follow your argument.

Reply: Thanks.

line 466: This is right, and this is what I mean with my specific comment 1). Only the present simulations do not show this instability because the SSW is too weak. Please, mention this because it is important when discussing observations.

Reply: It is mentioned in the conclusion part in the revision. In the future, WACCM will be utilized to further investigate the influence of SSW on QTDW under more realistic atmospheric conditions.

Please also note the supplement to this comment:
http://www.atmos-chem-phys-discuss.net/acp-2015-982/acp-2015-982-AC1-supplement.pdf

**Supplement:**

[revised manuscript text omitted]

Figure 6 shows the temporal variations of the W3 in meridional wind at  $\sim 9082$ km for case 1, case 2 and case 3. Note that the same perturbations for W3 were forced at the lower model boundary for all the three experimental runs. The W3 forcing was gradually increased from day 1 to 10, and was reduced after day 25 with constant amplitude between day 10 and 25. The perturbations of SPW1 in case 2 were nearly three times larger than case 3, both of which were sustained after day 10 with a Gaussian-shaped increase from day 1 to 10. The W3 in case 1 is the strongest with an amplitude of ~60 m/s (Figure 6a). The maximum amplitudes of the W3 in case 2 and
case 3 are ~40 m/s and ~35 m/s (Figure 6b and 6c), respectively. It is evident that the
amplitudes of the W3 are weakened during the SSW periods. In the following, we will
examine possible causes of the QTDW decrease during SSW.

The refractive index *m* of a forced planetary wave is [*Andrews et al.*, 1987]:

265
$$m^{2} = \frac{q_{\varphi}}{a(\overline{u} - c)} - \frac{s^{2}}{(a \circ \varphi)^{2}} - \frac{f^{2}}{4N^{2}H^{2}}, \qquad (1)$$

where *s*, *c*,  $\bar{u}$ , *a*,  $\varphi$ , *f*, *N*, and *H* are the zonal wavenumber, phase speed, zonal mean zonal wind, earth radius, latitude, Coriolis parameter, Brunt-V **ä**s **ä**l **ä** frequency, and scale height, respectively. Besides, And  $\bar{q}_{\varphi}$  is the latitudinal gradient of the quasi-geostrophic potential vorticity:

270
$$\bar{q}_{\varphi} = 2\Omega c \circ \varphi - \left(\frac{(\bar{u}c \circ \varphi)_{\varphi}}{a c \circ \varphi}\right)_{\varphi} - \frac{a}{\rho} \left(\frac{f^2}{N^2} \rho \bar{u}_z\right)_z, \qquad (2)$$

where  $\Omega$  is the angular speed of the earth's rotation,  $\rho$  is the background air density, and z means the vertical gradient. A necessary condition for baroclinic/barotropic instability is  $\bar{q}_{\varphi} < 0$ , and the planetary waves are propagating (evanescent) where  $m^2$ is positive (negative). Moreover, the meridional and vertical components (EPY and EPZ) of the Eliassen-Palm (EP) flux vector (F) for planetary waves can also be calculated with reconstructed wave perturbations from the TIME-GCM, defined following *Andrews et al.* [1987] as:

278
$$\vec{F}_{EP} = \begin{bmatrix} EPY \\ EPZ \end{bmatrix} = \rho a \cos \varphi \begin{bmatrix} \overline{u_z v' \theta'} & -\overline{v' u'} \\ \overline{\theta_z} & -\overline{v' u'} \\ \begin{bmatrix} r & (\overline{u} \cos \varphi)_{\varphi} \\ \overline{a} \cos \varphi \end{bmatrix} \frac{\overline{v' \theta'}}{\overline{\
[revised manuscript text omitted]

---

## Author Comment (AC2) · 15 Mar 2016

Interactive comment on "Influence of the sudden stratosphere warming on quasi-2 day waves" by S.-Y. Gu et al.

Anonymous Referee #2

General comments:

The authors use the thermosphere-Ionosphere-Mesosphere-Electrodynamics General Circulation model to model the effect of the sudden stratospheric warming on the quasi-2-day wave (QTDW). They investigate the non-linear interaction of the QTDW with westward zonal wavenumber 3 (W3) and the stationary planetary wave with zonal

wavenumber 1 and show that a QTDW with westward zonal wavenumber 2 can be produced.

Specific comments:

I agree with the specific comments of referee nr. 1 in addition I was wondering:

1) Why has the analysis of the W3 wave been performed in the meridional wind at 82 km, 7.5 S and days 25-30

Reply: We thank the reviewer for catching this. It should be 90 km and 22.5S. We changed some plots during the manuscript preparation without changing all the corresponding text. The mistakes are corrected in the revision. Figure 5b shows that the meridional perturbations of W3 maximize at ~90 km and 22.5°S in our simulations. Figure 6 shows that the W3 maximizes at around day 25-30 in all the three cases. Thus, we choose the peak location and peak time during the data analysis. 90 km and 22.5°S are also the grid points in TIME-GCM with no specific meanings.

2) Why are days 15-20 chosen for the analysis of the W2 and not the same time period as for the analysis of the W3?

Reply: In our simulations, W2 peaks earlier than W3 with maximum amplitude at around day 15-20, which is shown by Figure 10. When examining the vertical and global structure of W2 and W3, we would like to choose the periods with strongest wave perturbations.

3) Why has the analysis of the W2 wave been performed in the meridional wind at 100 km, 2.5 N and during days 15-20?

Reply: Figure 9b shows that the meridional perturbations of W2 maximize in the equatorial regions at ~100 km. We choose the peak location during the wavenumber-period spectrum analysis. 2.5N is also just one grid point near the equator in TIME-GCM.

4) In the caption of Figure 5 the authors state that the analysis of the W3 has been

performed at 22.5S and 90km. However, in the text describing Figure 5 (page 11, line 226f), the authors state that the analysis of the W3 wave has been performed at 82 km and at 7.5S. Which coordinates have been used?

Reply: We changed some plots during the manuscript preparation without changing all the corresponding text. It should be 90 km and 22.5S. The mistakes are corrected in the revision.

Technical corrections:

1) Page 15, Line 313: Baratropic/baraclinic => barotropic/baroclinic

Reply: Revised in the revision.